# rs66651343 and rs12909095 confer lung cancer risk by regulating *CCNDBP1* expression

Qiang Shi[1☯], Ji Ruan[2☯], Yu-Chen Yang[1], Xiao-Qian Shi[1], Shao-Dong Liu[1], Hong-Yan Wang[1], Shi-Jiao Zhang[1], Si-Qi Wang[1], Li Zhong[1]*, Chang Sun[1]*

1 College of Life Sciences, Shaanxi Normal University, Xi'an, Shaanxi, P. R. China, 2 State Key Laboratory for Conservation and Utilization of Bio-Resources in Yunnan, Yunnan University, Kunming, Yunnan, P. R. China

☯ These authors contributed equally to this work.
* sunchang@snnu.edu.cn (CS); lizhong@snnu.edu.cn (LZ)

**Data Availability Statement:** All relevant data are within the paper and its Supporting information files.

**Funding:** This work was supported by the Fundamental Research Funds for the Central Universities (2018CBLY005 [to QS], SY20210003

## Abstract

Lung cancer is a malignant tumor with high rates of mortality and shows significant hereditary predisposition. Previous genome-wide association studies suggest that rs748404, located at promoter of *TGM5* (transglutaminase 5), is associated with lung carcinoma. By analysis of 1000 genomes project data for three representative populations in the world, another five SNPs are identified to be in strong linkage disequilibrium with rs748404, thus suggesting that they may also be associated with lung carcinoma risk. However, it is ambiguous about the actually causal SNP(s) and the mechanism for the association. Dual-luciferase assay indicates that the functional SNPs are not rs748404, rs12911132 or rs35535629 but another three SNPs (rs66651343, rs12909095 and rs17779494) in lung cell. By chromosome conformation capture, it is disclosed that the enhancer encompassing the two SNPs, rs66651343 and rs12909095, can interact with the promoter of *CCNDBP1* (cyclin D1 binding protein 1). RNA-seq data analysis indicates that *CCNDBP1* expression is dependent on the genotype of these two SNPs. Chromatin immunoprecipitation assay suggests that the fragments spanning rs66651343 and rs12909095 can bind with the transcription factors, cut like homeobox 1 and SRY-box transcription factor 9, respectively. Our results establish the connection between genetic variations at this locus and lung cancer susceptibility.

## Introduction

Lung cancer is the dominating cause of cancer death and one of the most serious threats to public health [1]. The predisposing factors of lung cancer can be divided into genetic susceptibility [2] and many environmental ones, including cigarette smoking [3], ambient air pollution [4], asbestos or radon gas exposure [5, 6] and miscellaneous risk factors [7]. To explore the potential hereditary contribution for lung carcinoma, a lot of genome-wide association studies (GWAS) have been carried out (see GWAS catalog at https://www.ebi.ac.uk/gwas/ for detail), and one SNP at 15q15.2, rs748404, is proposed to be associated with lung cancer in Caucasian [8, 9]. Since rs748404 is located at the promoter region of *TGM5* (transglutaminase 5), it is suggested that rs748404 might be associated with lung carcinoma by regulating *TGM5* expression

[to LZ] and GK202001004 [to CS]) and National Natural Science Foundation of China (No. 31370129) to CS. The funders had no role in study design, data collection and analysis, decision to publish, or preparation of the manuscript.

**Competing interests:** The authors have declared that no competing interests exist.

[8]. In contrast, this association is hypothesized to be through the function of *TP53BP1* (tumor protein p53 binding protein 1) due to the position (~226.1 kb away from rs748404) and the role of this gene in carcinogenesis [9]. However, the potential molecular mechanism has never been investigated. Moreover, since rs748404 is within a long linkage disequilibrium (LD) block containing multiple SNPs [9], the true causal SNP(s) for lung carcinoma may be not rs748404 but other SNP(s) in this block.

In the present research, we sought to uncover the functional variations at 15q15.2 resulting lung carcinogenesis and underlying mechanism. The genotype data for rs748404 surrounding region was retrieved from 1000 genomes project (1000GP) and five SNPs were identified to be in strong LD with rs748404. Among them, three SNPs can affect enhancer activity in lung cell. Chromosome conformation capture (3C) assay was utilized to disclose the regulatory target gene. Our findings construct the contact between variants at 15q15.2 and lung carcinoma.

## Materials and method

### 1000GP data analyzing

The genotype neighboring rs748404 (250 kb upstream and downstream genomic region) was retrieved for three major representative populations in the world, CEU (Utah Residents with European Ancestry), CHB (Han Chinese in Beijing) and YRI (Yoruba in Ibadan), from the 1000GP public dataset (http://www.internationalgenome.org/). The LD pattern was calculated by ldSelect [10] or Genome Variation Server (http://gvs.gs.washington.edu/GVS150/). When $r^2 \geq 0.80$, the SNPs were supposed to be in strong LD.

### Genomic DNA extraction

Genomic DNA was isolated from Beas-2B (human lung/bronchus epithelial) cell line by phenol-chloroform approach. DNA quality was assessed by NanoDrop™ One (Thermo Fisher Scientific, Waltham, MA) and 1.5% agarose gel electrophoresis (Biowest Agarose G-10, Chai Wan, Hong Kong).

### Plasmids construction and Dual-luciferase reporter gene assay

By the primers listed in S1 Table, the GWAS tag SNP rs748404 neighboring regions (~1.5 kb) were amplified utilizing PCR with Phusion High-Fidelity DNA Polymerase (Thermo Fisher Scientific). The PCR product and pGL3-basic plasmid (Promega, Madison, WI) were digested with *Bgl*II and *Sma*I (NEB, Ipswich, MA), and then ligated by T4 DNA ligase (NEB). Three potential enhancers (rs35535692, rs17779494, rs66651343 and rs12909095 adjacent regions; ~1.5 kb for each) were amplified with the same method described above and primers in S1 Table. After *Kpn*I and *Mlu*I digestion (NEB), the segments were inserted into pGL3-promoter vector (Promega). Q5 Site-Directed Mutagenesis Kit (NEB) was utilized to produce the plasmid containing the corresponding allele for each SNP with the primers in S2 Table. All recombinant plasmids were sequenced to rule out any PCR errors and verify the orientation of the haplotypes before transfection.

Beas-2B or A549 (human lung cancer cell line) were cultured in Dulbecco's modified Eagle's medium (high glucose, HyClone, Logan, UT) with 10% fetal bovine serum (FBS, Biological Industries, Cromwell, CT) and 1% penicillin-streptomycin solution (Solarbio, Beijing, China) and incubated in 5% $CO_2$ at 37˚C. Cells (approximately $1.0 \times 10^4$ cells/well) were seeded into 24-well plates and cultured 24 h before transfection. 475 ng recombinant plasmid and 25 ng pRL-TK (Promega) were co-transfected into Beas-2B cells by utilizing Lipofectamine® 2000 (Thermo Fisher Scientific) according to the manufacturer's recommendation.

After 36 h culture, cells were lysed by Passive Lysis Buffer (Promega). GloMax Navigator (Promega) was utilized to measure the luciferase activity by Dual-Luciferase Reporter Assay System (Promega) according to the manufacturer's protocol. The relative luciferase activity was expressed as the ratio between firefly and *Renilla* luciferase. Six independent replicates were carried out for each experiment.

### 3C

3C was utilized to decipher the spatial genome organization between distal enhancer and promoter as previously reported [11]. In brief, formaldehyde (1% final concentration) was utilized to crosslink Beas-2B or A549 cells (approximately $10^8$). After lysing by Lysis Buffer, the crosslinked chromatin DNA was digested by *Hind*III enzyme (NEB) and ligated by T4 DNA ligase (NEB). The 3C library was extracted by phenol-chloroform method and DNA quantification was performed in Qubit® 3.0 fluorometer (Thermo Fisher Scientific).

Meanwhile, bacterial artificial chromosome (BAC) containing partial 15q15.2 region, RP11-1012I24 (BACPAC Genomics, Richmond, CA), was cultured for 16 h with 280 rpm shaking at 37°C and extracted by Large-Construct Kit (Qiagen, Valencia, CA) following the manufacturer's introduction. After digestion by *Hind*III, the BAC was ligated and purified by the abovementioned method.

The relative enrichment of 3C product was assessed by quantitative PCR (qPCR) with iQ SYBR green (Bio-Rad, Hercules, CA) and primers in S3 Table. The formula $2^{-\Delta\Delta Ct}$ was utilized to quantify the relative enrichment for chromatin. Three replicates were executed for each primer pair. The qPCR products were verified by resequencing.

### RNA-seq analysis

The RNA-seq data (sra format) for CEU lymphoblastoid cell lines (LCL) [12] was obtained from Sequence Read Archive database (https://www.ncbi.nlm.nih.gov/sra/) and converted into fastq format by NCBI SRA Toolkit (https://github.com/ncbi/sra-tools). After alignment with *CCNDBP1* (cyclin D1 binding protein 1) mRNA sequence by Bowtie2 [13], the expression was calculated by eXpress [14] with default parameter and reported in FPKM (Fragments Per Kilobase of transcript per Million fragments mapped) unit. The genotypes for LCLs were obtained from 1000GP or HapMap and linear regression analysis was performed between rs12909095 genotype and *CCNDBP1* expression level utilizing SPSS Statistics 20.0 (IBM, Armonk, NY).

### Chromatin immunoprecipitation (ChIP)

Potential transcription factors (TFs) were predicted by utilizing TRANSFAC database (http://www.gene-regulation.com/) and confirmed by ChIP assay with EZ ChIP Kit (MilliporeSigma, Burlington, MA) following the manufacturer's guidelines. In summary, the chromatin in Beas-2B or A549 cells (approximatively $10^7$) was crosslinked by formaldehyde (1% final concentration) for 15 minutes. After terminating by glycine for 10 minutes at 25°C, cells were washed twice with phosphate buffer saline (Solarbio) and collected into sterile centrifuge tubes (Eppendorf, Hamburg, Germany) by using a cell scraper. After lysing by SDS Lysis Buffer, the chromatin was sonicated by Ultrasonic Homogenizer (Scientz Biotechnology, Ningbo, China) to generate fragments (between 200 and 800 bp). The DNA/Protein complex was diluted using dilution buffer and precleared by protein G Agarose beads. Then, the normal mouse IgG and related mouse antibodies (Santa Cruz Biotechnology, Santa Cruz, CA) were utilized to immunoprecipitate DNA/Protein complexes. The precipitated protein/chromatin samples were washed by multiple buffers and resuspended into elution buffer. The Protein-DNA crosslink

was reversed and the protein was digested by proteinase K (Roche, Indianapolis, IN). DNA was purified by supplied spin columns and qPCR was utilized to quantify the enrichment with primers in S4 Table.

### Electrophoretic mobility shift assay (EMSA)

Beas-2B or A549 cells nuclear extracts were prepared by using Nuclear and Cytoplasmic Protein Extraction Kit (Beyotime, Shanghai, China) and assessed with Enhanced BCA Protein Assay Kit (Beyotime). The oligonucleotides for both alleles of rs66651343 and rs12909095 were synthesized by Sangon Biotech (Shanghai, China) and listed in S5 Table. After labeling with biotin by EMSA Probe Biotin Labeling Kit (Beyotime), the biotin-labeled duplex probes (10fmol) were incubated with the nuclear extracts (5 μg) for 20 min at 37˚C and electrophoresed on 4.9% non-denaturing polyacrylamide gel. After electrophoresis, probe-protein complex was transferred to positively charged nylon membranes (Beyotime). The probe-protein mixture on membranes were crosslinked by UV-light and incubated with streptavidin-HRP (horseradish peroxidase) conjugate (Beyotime). The EMSA images were captured in Luminescent Imaging Workstation (Tanon, Shanghai, China) by Chemilu-minescent EMSA Kit (Beyotime) following the manufacturer's protocol. For each reaction, the biotin-labeled duplex probes and probe-protein complex incubating with competitor probes (unlabeled duplex oligonucleotide, approximately 100-fold excess) were included as controls.

### Statistics

Independent Student's *t*-test was carried out to evaluate the different luciferase expression among recombinant plasmids, and the relative enrichment about 3C and ChIP by utilizing SPSS 20.0 Statistics. When $P<0.05$, the null hypothesis was rejected. The work was granted an exemption from requiring ethics approval by Ethics Committee of Shaanxi Normal University since no human subjects were involved.

## Result

### Genetic variations surrounding rs748404

Within the 500 kb region neighboring rs748404, there are 543, 636 and 969 SNPs for CEU, CHB and YRI populations, respectively. Among them, only five (rs35535692, rs12911132, rs66651343, rs12909095 and rs17779494) show (nearly) complete LD with rs748404 in CEU population (see S6 Table). These six SNPs constitute two distinct haplotypes (see Table 1) and haplotype 1 presents a high frequency (~76%) in CEU population. Considering the strong LD between these five SNPs and rs748404, these five SNPs should also present different

**Table 1. SNPs in the core haplotypes.**

| SNP ID | Position in chr15[a] | Haplotype 1 | Haplotype 2 |
|---|---|---|---|
| rs35535692 | 43544235 | C | T |
| rs66651343 | 43558004 | A | G |
| rs12909095 | 43558142 | A | G |
| rs748404 | 43559231 | T | C |
| rs12911132 | 43560436 | C | T |
| rs17779494 | 43579223 | C | T |

[a]Relative to human genome build 37.

distributions between lung cancer case and control groups in Caucasian. Another two SNPs, rs504417 and rs11853991, are suggested to be in strong LD with rs748404 in previous cohort [9]. However, these two SNPs only present a moderate LD with rs748404 in CEU ($r^2$ = 0.640 and 0.655). We further investigated the LD among these three SNPs in another four 1000GP populations from Caucasian, FIN (Finnish in Finland), GBR (British in England and Scotland), IBS (Iberian Population in Spain) and TSI (Toscani in Italia) and observed a similar pattern (results not shown). Therefore, these two SNPs are not included in our following functional genomics work. In CHB, only three SNPs, rs66651343, rs12911132 and rs17779494, show strong LD with rs748404 (see S6 Table). In contrast, none of the five SNPs present a high LD with rs748404 in YRI (see S6 Table).

### Function of the six SNPs

Since rs748404 and rs12911132 are located at *TGM5* promoter region (-184 and -1389 bp relative to the translational start site, respectively), we hypothesized that these two SNPs might alter promoter activity and further influence *TGM5* gene expression. Thus, we cloned the *TGM5* promoter region including these two SNPs into pGL3-basic plasmid, produced the plasmid containing another allele of these two SNPs by mutagenesis and transfected them into Beas-2B cell. For rs748404 and rs12911132, no significant differences are detected in luciferase activity between two alleles (*P* = 0.19 and *P* = 0.46, respectively; see Fig 1A), indicating that these two SNPs are not with the function to alter gene expression in lung tissue. To verify the

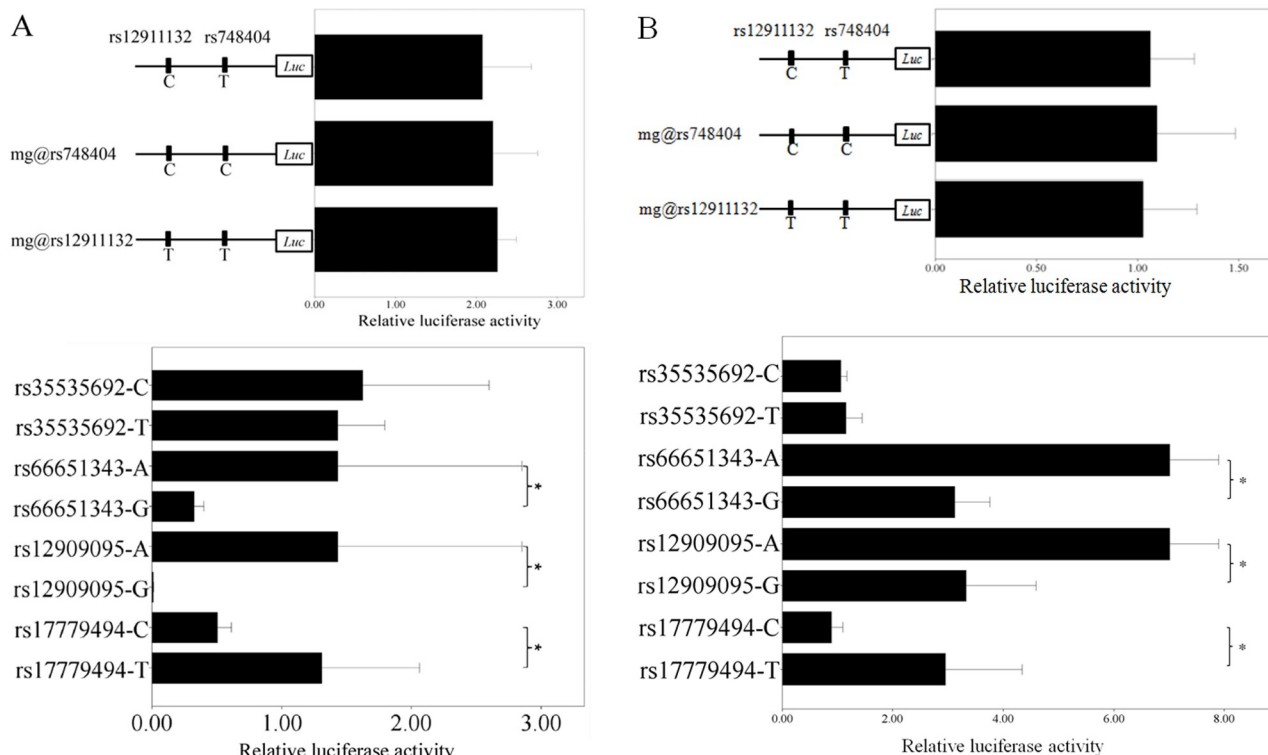

**Fig 1. Relative luciferase activity for different alleles of the six SNPs in Beas-2B (A) or A549 (B) cell.** For each part, the above chart is for promoter activity while the below one for enhancer. Each bar indicates one plasmid. For the two above charts, the above plasmid is the original construct, while the below two are from mutagenesis. Mg represents mutagenesis. For each SNP in below charts, the above allele is from haplotype 1 while the below one from haplotype 2. The *x* axis represents relative luciferase activity. All data is presented as mean ± standard deviation (SD) and normalized by the reading of empty vector. * represents *P*<0.01.

function of these two SNPs, we further transfected the plasmids into A549 and similar results are obtained ($P$ = 0.81 and 0.74 for rs748404 and rs12911132, respectively; see Fig 1B).

The remaining four SNPs (rs35535692, rs17779494, rs66651343 and rs12909095) are within the non-coding region and not located at the promoter of any identified genes. Therefore, we hypothesized that they might alter target gene(s) expression levels through enhancer model and evaluated the role of these four SNPs on gene expression regulation. Due to the long distance among these four SNPs (see Table 1), they were amplified and ligated into the vector separately (see S1 Table). For rs35535692, no significant difference is observed in luciferase activity between two alleles ($P$ = 0.39; see Fig 1A), indicating that rs35535692 is not with the function to alter gene expression. In contrary, G allele of rs66651343 and rs12909095 display ~77.3% and ~99.3% lower relative luciferase activity than A allele, respectively ($P$ = 0.0034 and $P$ = 0.0006, respectively; see Fig 1A). Meanwhile, C allele of rs17779494 presents ~61.4% lower luciferase activity than T one ($P$ = 0.0004; see Fig 1A), which indicated that these three SNPs are functional in lung cell and rs12909095 might play a more crucial role in expression regulation. The transfections into A549 yield the same pattern for these four SNPs ($P$ = 0.15, $8.43 \times 10^{-9}$, $3.86 \times 10^{-7}$ and $2.83 \times 10^{-5}$ for rs35535692, rs66651343, rs12909095 and rs17779494, respectively; see Fig 1B).

Moreover, there are multiple H3K4me1 and H3K27Ac peaks, two common histone modification around active enhancers [15, 16], nearby rs66651343, rs12909095 (see S1 Fig) and rs17779494 (see S2 Fig) in A549 cell. Considering their location and histone modification, we hypothesize that these three SNPs are locating at enhancers region and can alter enhancer activity. Since rs17779494 is locating as long as ~21.2 kb away from rs66651343 and rs12909095, they should belong to two different potential enhancers.

## Regulation target of these two enhancers

Despite of the fact that these three SNPs are within enhancer regions, it is still unclear which gene(s) they can regulate. To solve this issue, 3C was exploited to disclose the interaction between these two enhancers and nearby genes. Due to the long distance between the two enhancers, 3C was performed separately. The constant primers were set in the two enhancers, while the anchoring primers were set in the promoter of the four protein-coding genes (*CCNDBP1*, *EPB42* [erythrocyte membrane protein band 4.2], *TGM5* and *TGM7* [transglutaminase 7]) and ten random regions (see S3 Table).

As shown in Fig 2A, for the enhancer containing rs66651343 and rs12909095, increased ligation frequency is detected at promoters of *CCNDBP1*, *EPB42*, *TGM5* and *TGM7* (corresponding to 3th, 6th, 10th, 13th point in *x*-axis, ~76.1 kb, ~41.6 kb, ~1.7 kb and ~38.3 kb away from the enhancer, respectively) in Beas-2B cell. However, the reverse transcript (RT)-PCR with primers in S7 Table indicates that the mRNA expression of *EPB42*, *TGM5* and *TGM7* are not detected in Beas-2B (see Fig 3) and A549 (results not shown) cell. When these three genes are removed from analysis, only *CCNDBP1* promoter presents a strong interaction frequency in our assay (see Fig 4A). We further compared the ligation frequency between *CCNDBP1* promoter and other regions by utilizing one-sample *t*-test and a significant deviation can be detected ($P < 10^{-6}$), thus suggesting that *CCNDBP1* should be the regulation target of the enhancer containing rs66651343 and rs12909095 in Beas-2B cell. The 3C assay in A549 generates a similar pattern (see Fig 4B).

For the enhancer containing rs17779494, the 3C result is shown in Fig 2B and no significant difference is observed between *CCNDBP1* promoter and other genome regions ($P$ = 0.151). Therefore, the target gene of the enhancer containing rs17779494 remains unclear and deserves further investigation.

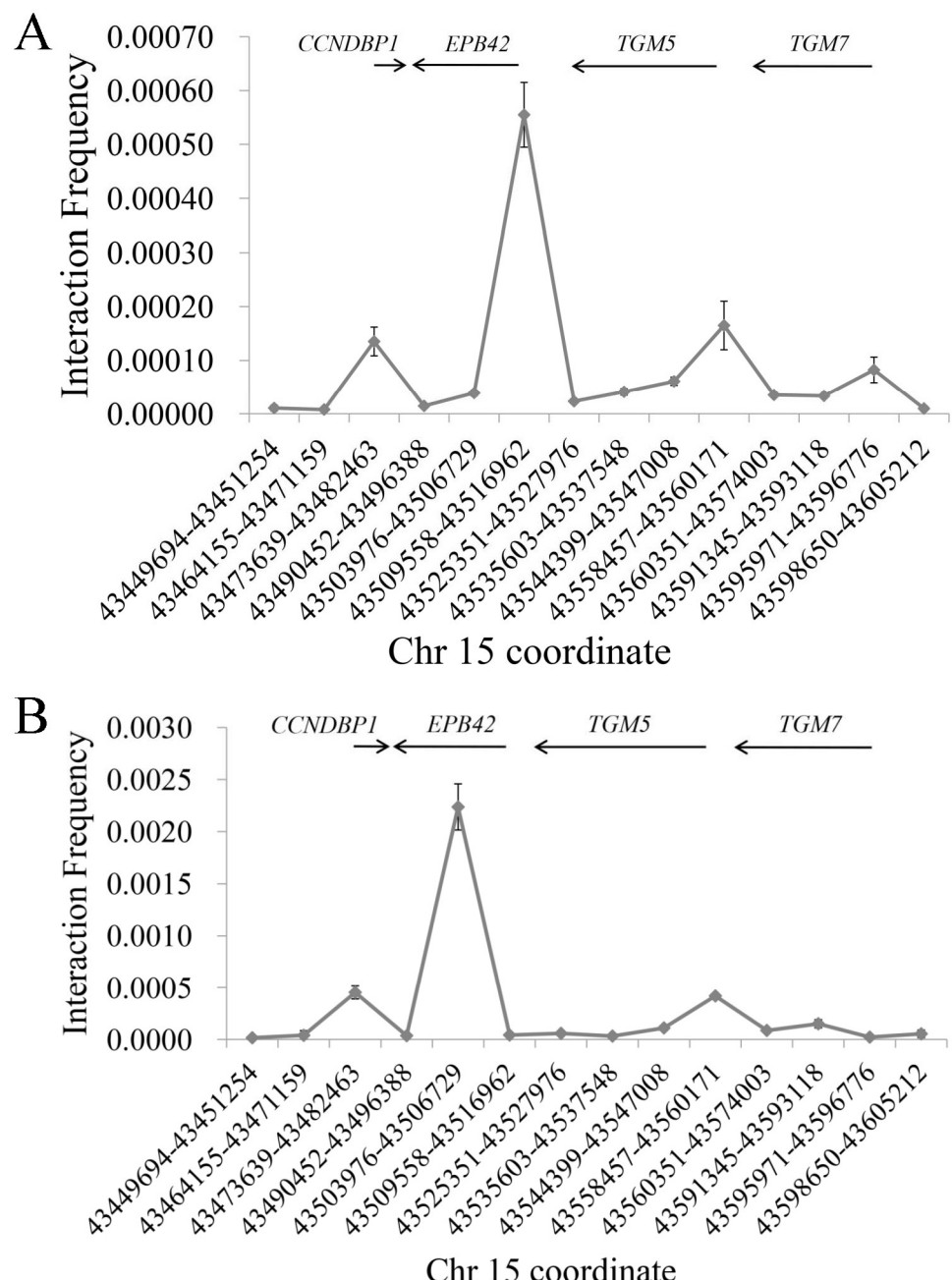

**Fig 2. Interaction efficiency in Beas-2B cell between the enhancer containing rs66651343 and rs12909095 (A) or rs17779494 (B) and surrounding genome regions in 15q15.2.** The *x* axis shows the start and end of the restrictive segments in chr15 (relative to human genome build 37) while the *y* axis indicates the relative interaction efficiency. The above arrow shows the schematic position and transcript direction of the gene in this region. All data is shown as mean ± SD.

## Association between rs12909095 genotype and *CCNDBP1* expression

If these two SNPs, rs66651343 and rs12909095, can indeed influence *CCNDBP1* expression, they should be expression quantitative trait locus (eQTL) for this gene. Since lung tissues with known genotype and expression are not available for us, a well-established model for eQTL

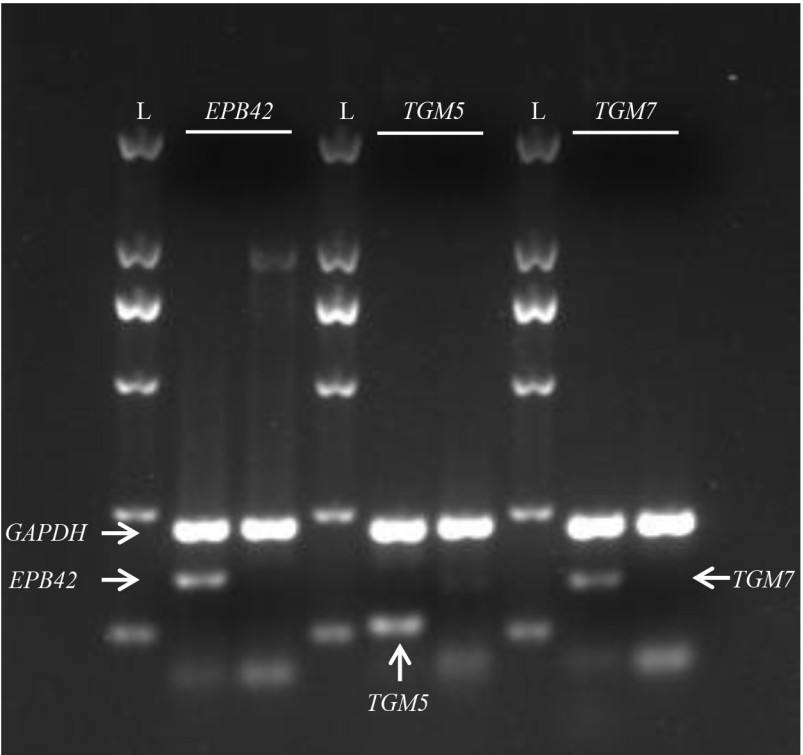

**Fig 3. Gene expression analysis (RT-PCR) for *EPB42* (left), *TGM5* (middle) and *TGM7* (right) in Beas-2B cell.** For each gene, the left lane is the result from positive control cell line while the right one from Beas-2B. The cell lines K562, HepG2 and SKOV-3 are used as a positive control for *EPB42*, *TGM5* and *TGM7*, respectively. L represents DNA ladder, which includes DNA fragments with known size (from top to bottom, 2000, 1000, 750, 500, 200 and 100 bp). *GAPDH* (glyceraldehyde-3-phosphate dehydrogenase) is utilized as internal control in PCR. The arrows point out the position of PCR product for different genes.

analysis, LCL, is utilized. We downloaded RNA-seq data for LCL [12] and calculated *CCNDBP1* expression. Since rs66651343 and rs12909095 are in complete LD in CEU ($r^2 = 1$), rs12909095 genotype was used to represent the haplotype. As shown in Fig 5A, rs12909095 genotype is significantly associated with *CCNDBP1* expression ($r = 0.252$, $P = 0.031$). More-over, A allele of rs12909095 is associated with higher expression, which is consistent with our luciferase result (see Fig 1). Similar results are obtained from GTEx Portal dataset (https://gtexportal.org/). The significant association between rs12909095 genotype and *CCNDBP1* expression is observed in multiple tissues, including brain-frontal cortex ($P = 0.0000028$), esophagus-muscularis ($P = 0.00030$) and cells-cultured fibroblasts ($P = 0.00038$; see Fig 5B). In all three tissues, A of rs12909095 is associated with higher *CCNDBP1* expression. Therefore, these two SNPs should be eQTL for *CCNDBP1*.

### TFs binding the two functional SNPs

Based on the fact that these two SNPs, rs66651343 and rs12909095, are located in enhancer, it seems that they might interact with TF and influence TF binding affinity. The prediction by Match at TRANSFAC indicates that rs66651343 and rs12909095 might be within the core binding region of CUX1 (cut like homeobox 1) and SOX9 (SRY-box transcription factor 9), respectively, and nucleotide substitution can induce the gain/loss of predicted TF. To confirm this prediction, ChIP was performed with related antibodies in Beas-2B cell and the relative

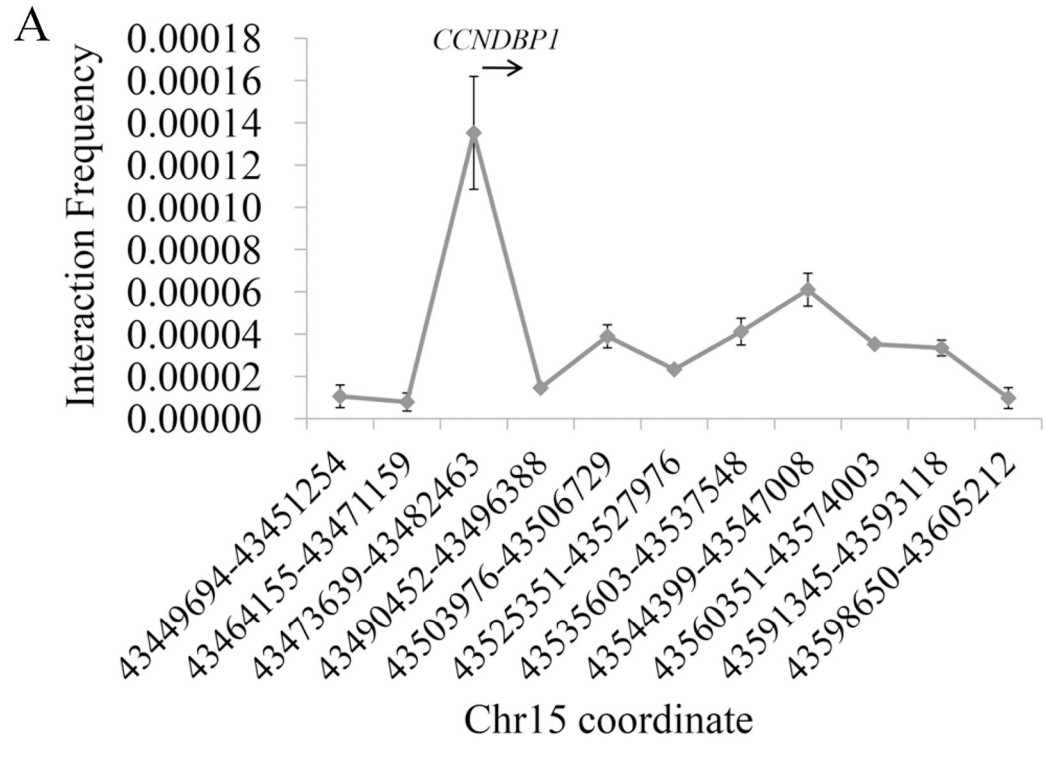

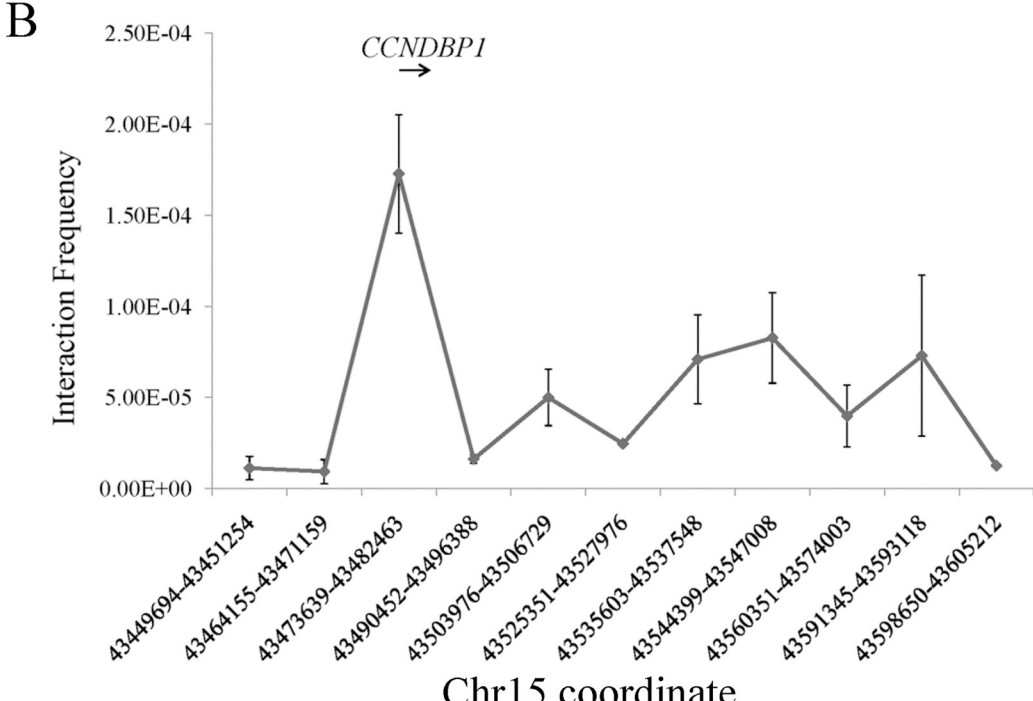

**Fig 4. Interaction efficiency in Beas-2B (A) or A549 (B) cell between the enhancer containing rs66651343 and rs12909095 and surrounding genome regions after removing three non-expressed genes.** The *x* axis indicates the restrictive segments location in chr15 (relative to human genome build 37) while the *y* axis shows the relative interaction efficiency. The above arrow shows the schematic position and transcript direction of the gene in this region. All data is shown as mean ± SD.

A

B

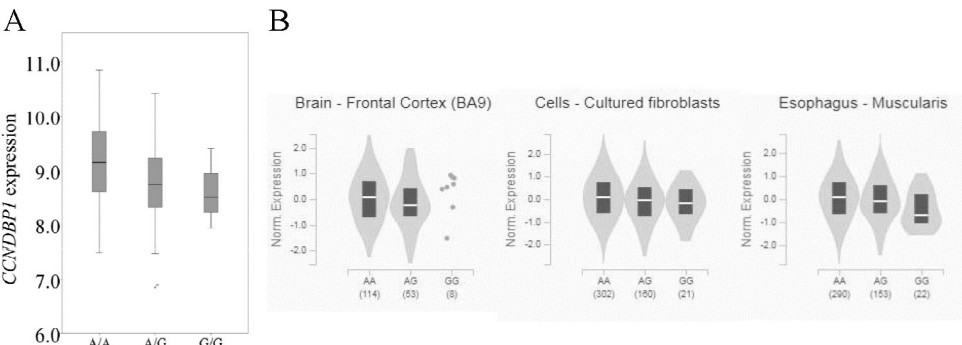

**Fig 5. Relationship between rs12909095 genotype and *CCNDBP1* expression in LCL at CEU population from literature [12] (A) and GTEx database (B).** The *x*-axis indicates genotype, while *y*-axis shows gene expression. For part A, the expression is displayed by FPKM and log transformed. For part B, the sample size for each group is shown in bracket.

chromatin enrichment was evaluated by qPCR. As shown in Fig 6A, compared with IgG, the chromatin surrounding rs66651343 and rs12909095 are significantly enriched by related antibodies (*P* = 0.013 and 0.038, respectively), thus confirming that these two TFs can bind the surrounding regions of the two SNPs in Beas-2B cell. The same conclusion can be obtained in A549 cell for rs66651343 and rs12909095 (*P* = 0.0088 and 0.032, respectively; see Fig 6B).

### Difference of TFs binding affinity between alleles

To examine the binding capacity differences between two alleles of rs66651343 and rs12909095, EMSA was carried out in Beas-2B cell. As shown in Fig 7A, the two alleles for rs66651343 and rs12909095 display apparently different affinity with nuclear proteins. Moreover, these patterns can be abolished by adding competitor oligonucleotides (see Fig 7A), which confirms that the binding between nuclear protein and probes are specific. The high expression allele, A of both SNPs, show a relatively low binding affinity. The similar pattern is also observed in A549 cell (see Fig 7B), which indicates that these two TFs might play a negative regulatory role in transcription.

### Discussion

In current research, population genetics and functional genomics approaches were utilized to investigate the real causal SNP(s) and the latent mechanism between genetic marker rs748404 and lung cancer predisposition. To achieve this goal, the 1000GP data were retrieved to analyze the LD pattern and five additional SNPs were identified to be in strong LD with rs748404. Further dual-luciferase assay showed that rs66651343 and rs12909095 can alter enhancer activity in lung tissue. Through 3C, the regulation target gene was identified to be *CCNDBP1*. Furthermore, the related TFs and molecular mechanism were uncovered. Our effort sheds more light on the mechanism between SNPs at 15q15.2 and lung cancer susceptibility.

It is interesting to observe that the tag SNP in GWAS, rs748404 [8, 9], is not with the function to influence gene expression in lung cell. This phenomenon is not surprising since rs748404 is chosen to represent this LD block in microarray not based on the function. Recent functional genomics work has frequently proposed that the causal SNP(s) for related phenotypes or diseases are not the tag SNP in microarry but the one(s) in LD with it [17–19].

Our reporter gene assay proposes that the functional mutations are rs66651343 and rs12909095, which is verified by eQTL analysis. The enhancer containing these two SNPs can form a loop with *CCNDBP1* promoter. All these facts suggest that these two SNPs can regulate

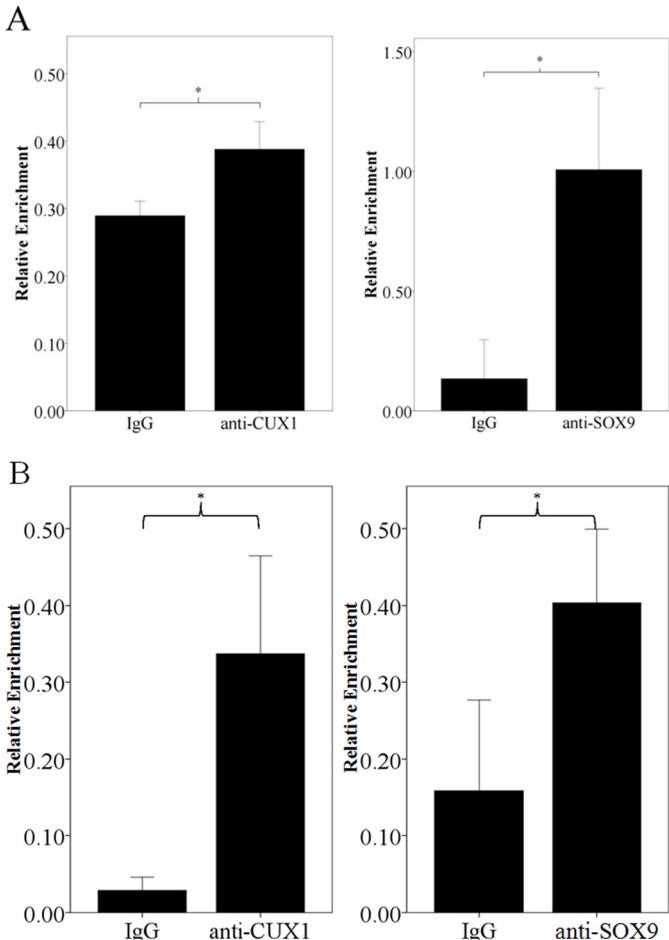

**Fig 6. Chromatin enrichment of the region containing position rs66651343 and rs12909095 in Beas-2B (A) or A549 (B) cell.** For each part, the left chart is for rs66651343 while the right one for rs12909095. The *y* axis represents relative enrichment. The result is normalized by input, and the data is presented as mean ± SD. * represents $P<0.05$.

the mRNA level of this gene. It is still unknown whether these two SNPs can influence the protein level. Further genome editing at this locus and protein quantification might clarify this issue.

Previous GWAS suggests that the SNPs in this locus may interact with *TP53BP1* to influence lung cancer susceptibility due to the importance of this gene in carcinogenesis [9]. Since the distance between *TP53BP1* promoter and the two functional SNPs, rs66651343 and rs12909095, is as large as ~244.7 kb, our 3C assay can not include this gene. Therefore, no conclusion can be made on this issue. However, there is no significant association between the SNPs at this locus and *TP53BP1* expression in GTEx database and our LCL analysis (results not shown), thus suggesting that *TP53BP1* is not likely to be the regulation target of the enhancer containing rs66651343 and rs12909095.

Instead, our 3C assay and eQTL analysis suggest that *CCNDBP1* is the regulatory target of the enhancer in lung cell. *CCNDBP1* is also known as *GCIP* (Grap2 and cyclin D-interacting protein), *DIP1* (D-type cyclin-interacting protein 1) or *HHM* (human homologue of maid), and encoding a ~1.3 kb mRNA transcript with ubiquitously expression in human tissues [20, 21]. CCNDBP1, composing of 360 amino acids with a calculated molecular mass of 40 kDa

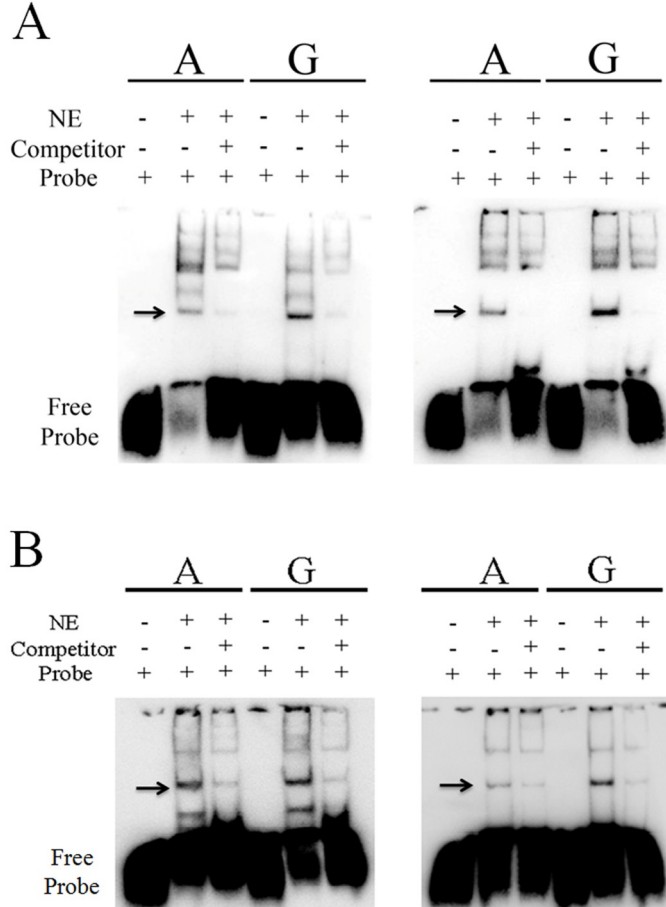

**Fig 7. Binding affinity difference of rs66651343 and rs12909095 alleles in Beas-2B (A) or A549 (B) cell.** For each part, the left chart is for rs66651343 while the right one for rs12909095. The top line indicates different alleles. NE denotes nuclear extracts, and the arrow points out the position of protein-probe complex.

and located mainly in the nucleus of cells, is involved in cell proliferation and differentiation [20, 22–24]. In lung cell, CCNDBP1 has been proposed to repress ID1 (inhibitor of DNA binding 1, HLH protein) expression, inactivate of PI3K/Akt signaling pathway and further suppress cell proliferation, migration, invasion and tumorigenesis, thus effecting as a tumor suppressor [25]. This function is strengthened by the fact that *CCNDBP1* expression is significantly higher in normal lung tissues than tumor ones [25]. Considering the role of rs66651343 and rs12909095 on *CCNDBP1* expression regulation, it can be concluded that these two SNPs can influence lung tumorigenesis by regulating *CCNDBP1* expression.

Besides lung, *CCNDBP1* has been proposed to be a tumor suppressor gene for multiple tissues, including prostate [26], breast [27–30], liver [30–33], ovary [34], lymph [35], retina [21], colon [36–38], neuroglial [39], bone [40] and lens epithelial cell [41] through interacting with different proteins. The significantly lower expression of *CCNDBP1* in cancer cell is appearing in almost all tissue types from TCGA (The Cancer Genome Atlas) project by utilizing UALCAN database (http://ualcan.path.uab.edu/index.html) [42] (results not shown). Moreover, the association between this locus and *CCNDBP1* expression is also observed in multiple tissues (see Fig 5). Considering all these facts, it might be hypothesized that rs66651343 and rs12909095 might influence carcinogenesis in multiple tissues, which deserves further investigation.

Our 3C assay identifies a special pattern in that the promoter of all four nearby genes are in relatively close space with the enhancer containing rs66651343 and rs12909095 (see Fig 2A), thus suggesting that these two SNPs might also regulate *TGM5*, *TGM7* and *EPB42* expression. This proposal is verified by the fact that these two SNPs are eQTL for these three genes in some tissues other than lung (see GTEx or Open Targets Genetics at https://genetics.opentargets.org/; results not shown). These facts indicate that this enhancer may be a shared one for nearby genes and effect through a tissue-specific manner. Therefore, these two SNPs may be associated with the diseases in which these three genes are involved.

Carboplatin and docetaxel are two types of anti-cancer agents for multiple solid tumors and acute leukemia and effect through inhibiting DNA biosynthesis, interfering cell growth and metastasis and inducing apoptosis of cancer cells *in vivo* [43–45]. Recent study has proposed that higher CCNDBP1 expression in lung cell can sensitize the chemotherapy of these two agents [25]. Considering the role of rs66651343 and rs12909095 on *CCNDBP1* expression regulation, these two SNPs may contribute to the different response of these two agent types among cancer patients, which deserves further investigation.

Our study has one limitation in that FBS, although widely used in tissue culture, might transform Beas-2B cell, which might further bias the result. To validate our conclusions, we investigated the function and effect of related SNPs in another lung line A549. The similar patterns between these two cell lines indicate that the potential Beas-2B cell transformation should not alter the *trans*-regulatory environment in cell and influence the function of related SNPs.

## Supporting information

**S1 Table. Primers in plasmid construction.**
(DOCX)

**S2 Table. Primers in mutagenesis.**
(DOCX)

**S3 Table. Primers in 3C-qPCR.**
(DOCX)

**S4 Table. Primers in ChIP-qPCR.**
(DOCX)

**S5 Table. Probes for rs66651343 and rs12909095 in EMSA.**
(DOCX)

**S6 Table. $r^2$ value between rs748404 and other SNPs in three representative populations.**
(DOCX)

**S7 Table. Primers in RT-PCR.**
(DOCX)

**S1 Fig. Histone modification for the segment surrounding rs66651343 and rs12909095 in A549 cell.** The yellow lines in left and right indicate the location of rs66651343 and rs12909095, respectively.
(TIF)

**S2 Fig. Histone modification for the segment surrounding rs17779494 in A549 cell.** The yellow line indicates the location of rs17779494.
(TIF)

**S1 Raw images.**
(PDF)

## Author Contributions

**Conceptualization:** Chang Sun.

**Funding acquisition:** Qiang Shi, Li Zhong.

**Investigation:** Qiang Shi, Ji Ruan, Yu-Chen Yang, Xiao-Qian Shi, Shao-Dong Liu, Hong-Yan Wang, Shi-Jiao Zhang, Si-Qi Wang.

**Supervision:** Li Zhong, Chang Sun.

**Writing – original draft:** Qiang Shi, Chang Sun.

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
