## [Decision Letter · Decision Letter 0]

16 May 2022

PONE-D-21-40574rs66651343 and rs12909095 confer lung cancer risk by regulating CCNDBP1 expressionPLOS ONE

Dear Dr. Sun,

Thank you for submitting your manuscript to PLOS ONE. After careful consideration, we feel that it has merit but does not fully meet PLOS ONE’s publication criteria as it currently stands. Therefore, we invite you to submit a revised version of the manuscript that addresses the points raised by the two reviewers during the review process.

We look forward to receiving your revised manuscript.

Kind regards,

Yan-Ming Xu

Academic Editor

PLOS ONE

**Journal requirements:**

**Reviewers' comments:**

Reviewer's Responses to Questions

**Comments to the Author**

1. Is the manuscript technically sound, and do the data support the conclusions?

Reviewer #1: No

Reviewer #2: Partly

2. Has the statistical analysis been performed appropriately and rigorously? 

Reviewer #1: No

Reviewer #2: Yes

3. Have the authors made all data underlying the findings in their manuscript fully available?

Reviewer #1: No

Reviewer #2: Yes

4. Is the manuscript presented in an intelligible fashion and written in standard English?

Reviewer #1: No

Reviewer #2: Yes

5. Review Comments to the Author

Reviewer #1: I have some questions for this manuscript:

1. The Beas-2B cells cultured with 10% FBS medium will transform, so the results are not reliable;

2. The figures are intricate with the simple labels and legends;

3. Could you show the mRNA or protein level of the gene regulated by the SNPs?

Reviewer #2: The current manuscript by Shi et al. has its merit by showing two SNPs, rs66651343 and rs12909095, are associated with lung cancer risk and may contribute to cyclin-D1-binding protein 1 expression. The experimental designs are pretty straightforward and scientifically valid in general. However, I do have one major concern: the authors used normal human lung BEAS-2B cells for many of their experiments, but FBS was used to culture the cells and this would have caused the transformation of the cell line – which may impact the results obtained. Therefore, all these results should be retested, and if possible, in multiple cell lines.

Other minor concerns/suggestions:

1. Unfriendly mode of English can be found.

2. Figures have to be more “compact”, e.g. the authors can merge a few of the main figures into 1 figure.

3. Many of the supplementary figures could become main figures (e.g., Figures S1, S4–S7).

4. The authors claimed that rs748404, a relatively well-characterized lung cancer risk-associated SNP, is not functional. This statement requires further verification and/or discussion (in both Results and Discussion sections).

6. PLOS authors have the option to publish the peer review history of their article (what does this mean?). If published, this will include your full peer review and any attached files.

Reviewer #1: No

Reviewer #2: No

---

## [Author Response · Author response to Decision Letter 0]

1 Sep 2022

Reviewer 1: 

The Beas-2B cells cultured with 10% FBS medium will transform, so the results are not reliable;

We verified our conclusions in another lung cancer cell line A549 and included the results. 

The figures are intricate with the simple labels and legends;

We revised the figure lables and legends. 

Could you show the mRNA or protein level of the gene regulated by the SNPs?

We talked about this issue in Discussion. 

Reviewer 2:

The authors used normal human lung BEAS-2B cells for many of their experiments, but FBS was used to culture the cells and this would have caused the transformation of the cell line – which may impact the results obtained. Therefore, all these results should be retested, and if possible, in multiple cell lines.

We verified our conclusions in another lung cancer cell line A549 and included the results. 

Unfriendly mode of English can be found.

We revised the text thoroughly. 

Figures have to be more “compact”, e.g. the authors can merge a few of the main figures into 1 figure.

We merged the previous luciferase, 3C and eQTL results into 1 figure. 

Many of the supplementary figures could become main figures (e.g., Figures S1, S4–S7).

We put the previous Fig S1 and S4-7 into main figures. 

The authors claimed that rs748404, a relatively well-characterized lung cancer risk-associated SNP, is not functional. This statement requires further verification and/or discussion (in both Results and Discussion sections).

We verified the conclusion in another cell line A549 and talked about this issue in Discussion.

---

## [Decision Letter · Decision Letter 1]

24 Nov 2022

PONE-D-21-40574R1rs66651343 and rs12909095 confer lung cancer risk by regulating CCNDBP1 expressionPLOS ONE

Dear Dr. Sun,

Thank you for submitting your manuscript to PLOS ONE. After careful consideration, we feel that it has merit but does not fully meet PLOS ONE’s publication criteria as it currently stands. Therefore, we invite you to submit a revised version of the manuscript that addresses the points raised during the review process.

We look forward to receiving your revised manuscript.

Kind regards,

Yan-Ming Xu

Academic Editor

PLOS ONE

Additional Editor Comments:

Reviewer 2 comments

Most of my concerns have been addressed, and the manuscript has been improved. However, I still have a few more minor suggestions/questions for the authors:

1. Data regarding A549 cells can be included in the main figures.

2. Similar results obtained with the A549 cells did not change the fact that FBS would transform BEAS-2B cells – this should be mentioned or discussed in the manuscript.

3. It was claimed that rs748404 was not a “functional” SNP, but why the short title of the manuscript stated: “association between rs748404 and lung cancer”?

4. English writing of the manuscript can be further polished.

Reviewers' comments:

Reviewer's Responses to Questions

**Comments to the Author**

1. If the authors have adequately addressed your comments raised in a previous round of review and you feel that this manuscript is now acceptable for publication, you may indicate that here to bypass the “Comments to the Author” section, enter your conflict of interest statement in the “Confidential to Editor” section, and submit your "Accept" recommendation.

Reviewer #2: All comments have been addressed

2. Is the manuscript technically sound, and do the data support the conclusions?

Reviewer #2: Yes

3. Has the statistical analysis been performed appropriately and rigorously? 

Reviewer #2: I Don't Know

4. Have the authors made all data underlying the findings in their manuscript fully available?

Reviewer #2: Yes

5. Is the manuscript presented in an intelligible fashion and written in standard English?

Reviewer #2: Yes

6. Review Comments to the Author

Reviewer #2: Most of my concerns have been addressed, and the manuscript has been improved. However, I still have a few more minor suggestions/questions for the authors:

1. Data regarding A549 cells can be included in the main figures.

2. Similar results obtained with the A549 cells did not change the fact that FBS would transform BEAS-2B cells – this should be mentioned or discussed in the manuscript.

3. It was claimed that rs748404 was not a “functional” SNP, but why the short title of the manuscript stated: “association between rs748404 and lung cancer”?

4. English writing of the manuscript can be further polished.

7. PLOS authors have the option to publish the peer review history of their article (what does this mean?). If published, this will include your full peer review and any attached files.

Reviewer #2: No

---

## [Author Response · Author response to Decision Letter 1]

1 Jan 2023

1. Data regarding A549 cells can be included in the main figures.

We have included A549 cell results in main figures. 

2. Similar results obtained with the A549 cells did not change the fact that FBS would transform BEAS-2B cells – this should be mentioned or discussed in the manuscript.

We added one paragraph in Discussion to talk about this issue. 

3. It was claimed that rs748404 was not a “functional” SNP, but why the short title of the manuscript stated: “association between rs748404 and lung cancer”?

We revised the short title as “Function of 15q15.2 SNPs in lung cancer”.

4. English writing of the manuscript can be further polished.

We invited a native speaker to revise the text and improve the language.

---

## [Decision Letter · Decision Letter 2]

21 Feb 2023

PONE-D-21-40574R2rs66651343 and rs12909095 confer lung cancer risk by regulating CCNDBP1 expressionPLOS ONE

Dear Dr. Sun,

Thank you for submitting your manuscript to PLOS ONE. After careful consideration, we feel that it has merit but does not fully meet PLOS ONE’s publication criteria as it currently stands. Therefore, we invite you to submit a revised version of the manuscript that addresses the points raised during the review process.

We look forward to receiving your revised manuscript.

Kind regards,

Yan-Ming Xu

Academic Editor

PLOS ONE

Journal Requirements:

Additional Editor Comments:

There are still minor issues to be addressed.

For example,

"Our study has one limitation in that FBS, although widely used in tissue culture, might transform Beas-2B cell, which might further bias the result. To validate our

conclusions, we investigated the function and effect of related SNPs in another lung cancer cell line A549. The similar patterns between these two cell lines indicate that

the potential transform should not alter the trans-regulatory environment in cell and influence the function of related SNPs."

Should be corrected as

"Our study has one limitation in that FBS, although widely used in tissue culture, might transform Beas-2B cell, which might further bias the result. To validate our

conclusions, we investigated the function and effect of related SNPs in another lung cell line A549. The similar patterns between these two cell lines indicate that

the potential Beas-2B cell transformation should not alter the trans-regulatory environment in cell and influence the function of related SNPs."

Before it can be formally accepted, the authors are required to perform thorough English grammar check and polishing for all the content throughout the article files.

Reviewers' comments:

Reviewer's Responses to Questions

**Comments to the Author**

1. If the authors have adequately addressed your comments raised in a previous round of review and you feel that this manuscript is now acceptable for publication, you may indicate that here to bypass the “Comments to the Author” section, enter your conflict of interest statement in the “Confidential to Editor” section, and submit your "Accept" recommendation.

Reviewer #2: All comments have been addressed

2. Is the manuscript technically sound, and do the data support the conclusions?

Reviewer #2: Yes

3. Has the statistical analysis been performed appropriately and rigorously? 

Reviewer #2: I Don't Know

4. Have the authors made all data underlying the findings in their manuscript fully available?

Reviewer #2: Yes

5. Is the manuscript presented in an intelligible fashion and written in standard English?

Reviewer #2: Yes

6. Review Comments to the Author

Reviewer #2: The manuscript has been improved. All my concerns have been addressed and I have no further comment.

7. PLOS authors have the option to publish the peer review history of their article (what does this mean?). If published, this will include your full peer review and any attached files.

Reviewer #2: No

---

## [Author Response · Author response to Decision Letter 2]

3 Mar 2023

Dear Dr. Yan-Ming Xu, 

Thank you for giving us the opportunity to revise our manuscript entitled “rs66651343 and rs12909095 confer lung cancer risk by regulating CCNDBP1 expression”. We have revised our manuscript according to the comments of the Editor and described our responses and the changes to the manuscript in detail in the attached sheet. All revisions have been marked as red. 

We hope you will find the manuscript acceptable for publication in PLoS ONE.

Sincerely,

Chang Sun 

Editor: 

"Our study has one limitation in that FBS, although widely used in tissue culture, might transform Beas-2B cell, which might further bias the result. To validate our conclusions, we investigated the function and effect of related SNPs in another lung cancer cell line A549. The similar patterns between these two cell lines indicate that the potential transform should not alter the trans-regulatory environment in cell and influence the function of related SNPs."

Should be corrected as

"Our study has one limitation in that FBS, although widely used in tissue culture, might transform Beas-2B cell, which might further bias the result. To validate our conclusions, we investigated the function and effect of related SNPs in another lung cell line A549. The similar patterns between these two cell lines indicate that the potential Beas-2B cell transformation should not alter the trans-regulatory environment in cell and influence the function of related SNPs."

We revised related text. 

Before it can be formally accepted, the authors are required to perform thorough English grammar check and polishing for all the content throughout the article files.

We revised the grammar and polished the text.

---

## [Editor Report · Decision Letter 3]

29 Mar 2023

rs66651343 and rs12909095 confer lung cancer risk by regulating CCNDBP1 expression

PONE-D-21-40574R3

Dear Dr. Sun,

We’re pleased to inform you that your manuscript has been judged scientifically suitable for publication and will be formally accepted for publication once it meets all outstanding technical requirements.

Kind regards,

Yan-Ming Xu

Academic Editor

PLOS ONE
---

## [Editor Report · Acceptance letter]

6 Apr 2023

PONE-D-21-40574R3 

rs66651343 and rs12909095 confer lung cancer risk by regulating *CCNDBP1* expression 

Dear Dr. Sun:

I'm pleased to inform you that your manuscript has been deemed suitable for publication in PLOS ONE. Congratulations! Your manuscript is now with our production department. 

Kind regards, 

on behalf of

Dr. Yan-Ming Xu 

Academic Editor

PLOS ONE